# Autoencoders Based on 2D Convolution Implemented for Reconstruction Point Clouds from Line Laser Sensors

**DOI:** 10.3390/s23104772

**Published:** 2023-05-15

**Authors:** Jaromír Klarák, Ivana Klačková, Robert Andok, Jaroslav Hricko, Vladimír Bulej, Hung-Yin Tsai

**Affiliations:** 1Institute of Informatics, Slovak Academy of Sciences, 845 07 Bratislava, Slovakia; 2Department of Automation and Production Systems, Faculty of Mechanical Engineering, University of Zilina, 010 26 Zilina, Slovakia; 3Department of Power Mechanical Engineering, National Tsing Hua University, Hsinchu 30013, Taiwan

**Keywords:** autoencoder, reconstruction, point cloud, scanning, Industry 4.0

## Abstract

Gradual development is moving from standard visual content in the form of 2D data to the area of 3D data, such as points scanned by laser sensors on various surfaces. An effort in the field of autoencoders is to reconstruct the input data based on a trained neural network. For 3D data, this task is more complicated due to the demands for more accurate point reconstruction than for standard 2D data. The main difference is in shifting from discrete values in the form of pixels to continuous values obtained by highly accurate laser sensors. This work describes the applicability of autoencoders based on 2D convolutions for 3D data reconstruction. The described work demonstrates various autoencoder architectures. The reached training accuracies are in the range from 0.9447 to 0.9807. The obtained values of the mean square error (MSE) are in the range from 0.059413 to 0.015829 mm. They are close to resolution in the Z axis of the laser sensor, which is 0.012 mm. The improvement of reconstruction abilities is reached by extracting values in the Z axis and defining nominal coordinates of points for the X and Y axes, where the structural similarity metric value is improved from 0.907864 to 0.993680 for validation data.

## 1. Introduction

Standard systems being applied in the industrial field are built primarily on the use of camera devices that create visual data classifiable as 2D data. With the gradual development and reduction of the price, more and more LiDAR-type laser sensors are applied for area surveys or outdoor sensing. Such applications can be seen primarily in connection with unmanned aerial vehicles (UAVs) [1,2], unmanned ground vehicles (UGV) [3,4], autonomous vehicles [5], or with household devices, such as robotic vacuum cleaners and the like [6]. At the same time, sensors for sensing physical quantities are being developed for industrial applications [7,8], which show the increasingly significant direction of development and the physical possibilities of the future development of various types of sensors including new body materials [9,10]. In the field of industry, not only point and line lasers are used for measurement and inspection systems. The advantage of these line lasers is the high accuracy for scanning surfaces, where the resolution in the Z axis ranges from 0.012–0.001 mm depending on the type of sensor [11]. The resolution in the X and Y axes depends on the hardware setting and scanning parameters [12]. It is possible to scan the inspected surface with a width of up to 10–270 mm, and the number of scanned points can be up to 2048 in one line, depending on the manufacturer and model of the sensor [11,13]. By using such sensors, it is possible to obtain high-quality data of the scanned surface. Another type of a laser device is the VR-6000 optical profilometer made by KEYENCE Corporation. This device is adapted for very accurate scanning of objects with a scanning accuracy of up to 0.0001 mm [14]. This predestines these sensors for application in inspection systems for industrial purposes [15]. In the field of inspection systems, the area of surface quality evaluation is mainly quantitative or qualitative assessment. Quantitative evaluation of scanned surfaces can be performed using statistical methods. In the field of qualitative evaluation of surfaces, this task is significantly more complicated. A large part of the issue is focused on the detection of defects on the surface of inspected objects. The problem of defect detection can be solved in two ways. One of them is direct detection of defects in visual data using the R-CNN [16], fast R-CNN [17], faster R-CNN [18], or YOLO [19] methods included in the supervised learning area. The second way is through the detection of anomalies by methods such as autoencoders, U-networks, visual transformers, and the like included in the area of unsupervised learning. Based on works [5,6,7,8,9], this is suitable for the initial identification of anomalies and the application of unsupervised learning methods in the first step. The field of visual data is relatively well processed and verified for the field of autoencoders. Applying these methods in the field of point cloud is still under development. This is primarily due to the emergence of high-quality sensor devices on the market. The second reason is the difference between typology of geometric data and classical visual data. This is the primary reason why the work with point clouds has not yet been fully developed and requires a relatively large amount of research and focus on its processing and evaluation. The illustration of basic inspection systems is shown in Figure 1. Two methods of defect detection and anomaly detection are mentioned there. The first method is based on detectors, which are trained to find trained patterns of defects which are bounded in regions and labeled. The main disadvantage is high sensitivity to the diversity of defect patterns. The second method is anomaly detection, where anomalies are defined as differences between reconstructed samples generated by trained reconstruction model (in this experiment by autoencoder) and tested samples. This type of approach highlights differences—anomalies. The disadvantage of this approach is not labeling the detected objects. In the papers already published, many types of systems are mentioned, based on one type of approach or combination of more approaches, and create hybrid systems. Each method has its own advantages and disadvantages. 

The majority of published works are focused on laser sensors of the LiDAR type. In connection with the implementation of autoencoders, the focus is primarily on indoor applications, where the emphasis is placed on the classification of the cloud of points representing a specific object and, subsequently, in the process of reconstruction, creating a sample cloud of points of the given object. The use of laser sensors in the inspection system is applied for the detection of defects on rails [20]. The principle is based exclusively on a constant sensed rail pattern. Thanks to this, it is possible to identify abnormalities or defects in the scanned surface. Threshold functions were used for this purpose. A similar system for the detection of defects using laser sensors on the surface of the rails was published in [21]. This work describes the use of regression and extraction of differences from the mathematical description of the shape scanned from above. In the case of more complex works, these methods cannot be used due to the various patterns occurring on the surface of the inspected object. In this way, it is more appropriate to ensure an ideal pattern of the controlled part of the scanned object. The work [22] describes the method of finding anomalies in visual content through the cascaded autoencoder (CASAE) architecture based on two levels of autoencoder. The result is a sample image generated, and when compared with image testing, the difference is in the identified anomalies. The development of an efficient quality of assessment method for 3D point clouds is described in [23]. There are described algorithms to improve and evaluate 3D point clouds assessment. 

One of the first works dealing with the recognition and classification of point clouds is focused on the tree solution of the neural network [24]. Working with point clouds consists of formatting 3D shapes represented by point clouds. These point clouds are arranged in one-dimensional directions. The processing of such data is based on 1D convolution. The goal is to create a representative pattern of the object based on the input cloud of points or an image that can be characterized by the generated cloud of points. The basis of this solution is similar to the PointNet architecture. This work describes the classification of the cloud of points into the appropriate category [25]. In this work, data are characterized as a set of points in 3D space, where each point is represented by a vector containing data such as the coordinates of a point in Euclidean space, a color channel, and the like. Another similar work is dedicated to the accuracy of point cloud reconstruction of 4 types of data, where the noise is monitored against the generated point cloud and against the input point cloud [26]. The basis of this architecture is built on a 1D arrangement of a cloud of points in Euclidean space. The architecture is built on 1D convolution. The number of points in point cloud is 2048. Another work is based on the processing of the point cloud through voxelization and transformation into a 3D object, for which 3D convolution is used [27]. A developing area is autonomous transport, where LiDAR sensors are used for scanning mainly exteriors. Autoencoders are also applied here for the reconstruction of this type of data [28]. The basis of the architecture is built on 1D convolutions. The aim of the work is to minimize the demands on memory and data storage. Another work in the field of application of autoencoders is described in [29]. The contribution of this work is to point out new perspectives in the processing of point clouds with emphasis especially on recorded large-scale 3D structures from standard data. The self-supervised model is described in [30], where the emphasis is on predicting the density of points in the point cloud database of the same category. Another work focused on data compression is described in [31]. The authors changed their view of the issue from global reconstruction to local reconstruction and thereby captured or focused on the internal structure of point clouds. Most recently, solutions built on transformers are applied in the works as shown in [32] with very good results achieved in this area. The application of autoencoders to increase or decrease the number of points in a dataset with an emphasis on the connection and combined application with sensing devices and CAD models is described as very desirable [33]. The application of the SSRecNet architecture with the exponential linear unit (ELU) activation function on the ShapeNet and Pix3D datasets is described in [34]. The work [35] describes a transformer called Point-BERT. The application of this model is compared with standard models, where the tests were performed on the ModelNet40 dataset, while the Point-BERT model achieved an accuracy of 93.80%. The autoencoders for point clouds or data from MRI and CT devices are applicated in medical areas. Autoencoders are used for these data to detect anomalies in the way of tumors, etc. [36].

The research described in this work is focused on reconstruction models—autoencoders based on 2D convolution and using high precision 3D data from the laser sensor. The goal of this work is to design a system adapted for generating a sample point cloud with respect to the tested point cloud. The purpose of our work is to design methods of reconstruction of point clouds obtained from laser sensors as very precise data and their use in inspection systems for anomaly detection by unsupervised learning. The novelty of this work is in the application of autoencoders based on 2D convolution to reconstruct 3D data with the ability to get very high similarity between tested and reconstructed samples, and, in this way, to use high-potential data quality from laser sensors. 

## 2. Point Cloud Reconstruction Based on 2D Convolution

Most of the works focus on the processing of the cloud of points with the aim of optimization, classification, or homogenization of data. On the basis of previous works and the application of inspection systems as in common applications, the aim of this work is to design a system capable of reconstructing an image with adequate quality as achieved by scanning procedures with laser sensors. These values are highly dependent on the type of device used and the scanning parameters. In general, the resolution in the Z axis is defined by the type of sensor. The resolutions in the X and Y axes are highly dependent on the type of sensor and the way of application. The resolutions in X and Y are also dependent on the scanning parameters such as distances between sensor and scanned surface, shape of surface and relative movement of sensor with respect to the scanned object. Laser sensors can achieve a resolution of 0.01 to 0.1 mm (scanCONTROL 2600–100 [11]) in the X or Y axes according to the shape of the scanned surfaces [12]. This work describes the system for reconstruction of data from these sensors using autoencoders in industrial applications with high precision data from the laser sensors. The reconstruction of data is based on usage of 2D convolution comparable to the use of autoencoders on visual data. The standard methods described in the articles on processing point clouds are built either by 1D (conv1D) or 3D convolution (conv3D) using voxelization (3D) [24,28,30,33]. The result is the design of a system including data acquisition, data processing, autoencoder training, and sample data generation using the autoencoder for encryption of the data and their comparison with the tested data, including the evaluation, as depicted in Figure 2. which illustrates a pipeline of data. The first step is the scanning procedure, which consist of a calibration process in the way of scanning the rotary shaft and removing the first harmonic component as eccentricity of assembling the rotary shaft to the rotary axis. This work is described in [37]. This way, the processed data are obtained, decrypted, and normalized to continuous values from 0 to 1, the most suitable for the training process. For illustration, decrypted values are transformed to discrete values from 0 to 255 in 3 channels, red, green and blue (RGB), as an image. Decrypted data are used for training autoencoders and reconstruction in trained model. Reconstructed data are obtained as encrypted data consisting of continuous values from 0 to 1. The reconstructed data are transformed to discrete values to get visual representation of reconstructed data in RGB (image). The reconstructed data are then encrypted, and the point cloud is obtained, which is afterwards possible to be compared with the processed data. The comparison of encrypted reconstructed data and processed data are in the form of evaluated data. The whole algorithms are designed in Spyder environment and implemented in Anaconda software [38].

### 2.1. Preprocessing Data

To generate a sample cloud of points using 2D convolution, it was necessary to prepare the data. In the raw state, the data from the laser sensor were obtained in the form of .csv files, where all scanned points were stored. From the point of view of the operation of the laser sensor, it is possible to format this point cloud into a 2D space—i.e., a matrix, where each point is represented by a vector of its coordinates (X, Y, and Z). Based on the X and Y coordinates of the points, the points in the matrix were divided into a discrete place in the matrix based on their X and Y values (Figure 3A) In this way, representative content can be obtained. Subsequently, it was necessary to homogenize the data and remove inaccuracies in the data that would worsen the results in the process of training the autoencoder or in the process of data reconstruction. The first data homogenization consisted of removing the first harmonic component in the data. This inaccuracy arises when the scanned object is placed incorrectly on the rotating axis of the shaft. This was reflected in the sinusoidal display of the scanned object. The process of compensation and removal of this harmonic component is described in [37] using a two-step approach, where in the first step, the first harmonic component with a phase shift was identified based on the Fourier transform. For the sake of accuracy, this method was supplemented with a custom method for correcting the phase shift and the frequency of the first harmonic component in the data. The data with the first harmonic component removed are shown in Figure 3B) The next step was to remove empty spaces due to non-scanned points by the laser sensor. These places were filled with the average value of the surrounding points based on Equation (1), where the average value was computed over non-zero values in a 3 × 3 matrix. The result was the generation of a cloud of points corresponding to the visual content shown in Figure 3C) The display of the monochromatic expression of the Z-coordinates of the scanned points is shown in Figure 3D) where the details of the scanned surface of the gear wheel can be better observed.
(1)$$V_{0}=\frac{\sum_{i.j}P_{i,j}}{n},\;P_{i,j}\neq 0,\;n>3$$

### 2.2. Architectures of Autoencoders

The key part of this work is focused on the architecture of the used autoencoder together with the type of data used. Based on the scanning parameters of the used laser scanner, which scans a maximum of 640 points in one line, this data were divided into fragments, while the range of the measuring band was preserved (dimension X, i.e., 640 points), and the dimension in the Y axis was experimented with. The size of the training data fragments was defined as 96, 120, 240, 360, 480, and 640. At the same time, the training process of different autoencoder architectures was carried out for each of the sizes. The original data consisted of 3 scans in the size of 640 × 11,000, where 2 scans were used to create the training dataset and the 3rd was used to create the verification dataset. In total, 4 basic types of autoencoders were defined (Table 1), where the simplest was built on 9 convolutional layers with the number of filters in the latent space at 128. More complex architectures were built on 512 filters in the latent space, and the overall summary of the architectures is given in Appendix B. The goal was to find a way for the system to be able to achieve high quality of the reconstructed image from data preparation through the architecture to the parameters of the training process. The main purpose was to explore and define the impact of specific parts of architecture in order to reach better results in reconstructing point clouds—for instance, the impact of a higher number of trainable parameters or usage of more parameters at the end of architectures or balanced architecture. The more accurate specification is defined in Table 1. The first type of architecture was created as basic architecture with 128 filters for each convolution layer without the input layer and last layer. The other architectures used more parameters in the way of more filters in convolution layers or implementing more convolution layers in specific parts of the architectures. For each type of architecture, there was a specific shape of input data tested, such as fragmented data (96, 120, 240, 360, 480, and 640). The architectures were built on the TensorFlow library [39]. The experiments were processed for four types of architectures, each modified to six different-sized training data inputs. The evaluation of designed architectures was important in two areas. The first area consists of achieved training results in the way of training accuracy and loss values. The second monitored area was the comparison of the error between the reconstructed and the input cloud of points together. The variables under consideration are the size of samples connected with their numbers. The amount of training data and number of parameters were dependent on the size of the graphics memory of 24GB for the RTX3090 GPU. For this reason, the most training data were in the dimension 640 × 96, and with increasing data size to the dimension 640 × 640, the number of batch size decreases. The optimizer used was Adam [40]. The loss function used was mean square error (MSE).

## 3. Results

The summary of the training results is presented in Table 2. The overall results are in Table A1 (Appendix A). From the view of the type of architecture, it can be said that the best results were achieved for ITE_1 with the least number of training parameters. With increased number of filters meaning more training parameters, the achieved results worsened. Additionally, with higher number of trainable parameters and higher number of convolution layers, training processes were very sensitive to overtraining. The size of input data is attached with number of training and validation samples, and there is an inverse relationship between the size of samples and the number of samples. It is due to scans fragmentation to specific sample sizes. The best results were therefore achieved with the dimensions of 640 × 96 (L_96_ITE_x), and good results were also achieved for the size of 640 × 120 (L_120_ITE_x) as shown in Figure 4 and Figure 5. In the case of data dimensions of 640 × 240 (L_240_ITE_x), 640 × 360 (L_360_ITE_x), 640 × 480 (L_480_ITE_x), and 640 × 640 (L_640_ITE_x), only average or below average results were observed. The summarization of training results is shown in Figure 4. The best accuracies and loss values achieved for each type of architecture and input size of samples are illustrated there. The second illustration of results, Figure 5, shows mean square errors with standard deviation of mean square errors. As it is shown that the best results from the view of architecture were achieved for the first architecture ITE_1 and for input size, the best results were achieved for the size of 940 × 96. In Figure 6, the structural similarity index (SSIM) defined in [41,42] is shown. According to this metric, the result 0.988568 for L_96_ITE_1 is very good, which indicates almost identical reconstructed point cloud to the original point cloud of samples. The complete results are shown in Appendix A in Table A1. The illustrations of point cloud (3D) are in Table A2. There are 3 types of images, where the first one is showing an original sample, the second image is a constructed sample, and the last one is a reconstructed point cloud showing difference values between the reconstructed image and the original sample. The visualizations of encoded and decoded images of point clouds in 2D are displayed in Table A3. The table shows visual representations of encoded point clouds and reconstructed point clouds. The last column shows differences between them. Among all the results, the best results were reached for architecture ITE_1 (Appendix A, Table A4).

Based on the results, it is assumed that due to the small amount of data, it is more appropriate to use a simple balanced architecture with a smaller number of filters. This statement is supported by the results especially for the ITE_1 architecture defined in Figure 4. Accuracy is decreasing in direct ratio with the number of training samples. For L_96_ITE_1 679 training samples were used and the achieved results have accuracy of 0.9807, loss 0.0008 and structural similarity metric 0.988568 in Figure 6 (more specific in Table A1). In case of L_640_ITE_1 97 training samples were used. This way results accuracy decreased to 0.9669, loss is 0.0019 and structural similarity metric is 0.944652 (more specific in Table A1). The metric of mean square error supports also this statement defined in Figure 5. The similar tendencies are for other architectures but not to be represented so clearly. For further work the architecture L_640_ITE_1 is used.

There is a possibility for the structural similarity metric improvement, which can be made by focusing on the values for individual axes. It can be demonstrated on the validation sample. The X and Y axes can be characterized as the positions of the scanned points and considered as constants. Values representing the positions of the points in space are essential for determining the position on the Z axis. It is due to principle of obtaining data from the laser sensor. For this purpose, it is convenient to divide the coordinates of points into 3 matrices for each axis and to express the deviation of the reconstructed points from the reference points. The expression of these values is given in Table 3. In this table, there is a graphic representation of the deviations for individual axes and the return of the average error for a specific axis. The mean square error for the entire improved frame is 0.136640 mm. The mean square error of unmodified validation sample is 3.488547 mm. The value SSIM improved from 0.907864 to 0.993680. This illustrates how it is possible to suppress the error of the reconstructed image and thereby increase the accuracy of the reconstruction of the point cloud. A graphical representation of the application of the extraction of only the Z-coordinates of the reconstructed points in conjunction with the nominal coordinates of the points for the X and Y axes is shown in Figure 7.

## 4. Conclusions

The goal of this work is to create a methodology for creating inspection systems based on 3D data obtained by using laser sensors. Most of the work focuses on standard point cloud datasets. The basic part of autoencoders is built usually on 1D convolution or 3D convolution often associated with voxelization. In this paper, we present the possibilities of working on the reconstruction of the point cloud based on 2D convolution on very accurate data from laser sensors. The goal is to provide a more comprehensive overview of the creation and training of such autoencoders as well as the impact of the type of training data on the accuracy of the reconstructed data compared to the input data. Another feature is to focus on smaller dimensions, namely 640 × 96 and 640 × 120, where better results are achieved but where it is also possible to reconstruct the points representing the stamped numbers as the description of the gear wheel. The architecture of the autoencoder itself does not have such a significant effect. The accuracy is connected to the topology of architecture and the amount of data. This is based on the results shown in Figure 4, mainly for architecture ITE_1, where there is an indirect ratio. For other architectures, there are the same tendencies but without such obvious results. MSE 0.015829 mm and SSIM 0.988568 were reached. The validation was performed on validation data independent from training data, where MSE 3.488547 mm and SSIM 0.907864 were reached. In the improved method, we reached MSE 0.136640 mm and SSIM 0.993680. The results show that the accuracy of autoencoders is almost comparable to the resolution of laser sensors, meaning very good results. The performance of training is highly sensitive to the amount of data according to the number of layers and the number of filters used in the architecture. Another outcome is the possible applicability and possibility for further development of autoencoders based on 2D convolution for point cloud processing.

## 5. Discussion

In the experiments, there were demonstrated possibilities of samples reconstruction by trained autoencoders. The results have shown that in case of data from laser sensors, it is possible to reach very good results—for instance, SSIM 0.993680. The limiting factor was observed in the amount of data and GPU memory (Nvidia Gforce RTX3090, 24 GB memory) mainly in the case of a training dataset with bigger size—for instance, 640 × 480 and more—and the number of filters in convolution layers. A necessity for further development is the provision of server graphics cards (GPU) with larger graphics memory due to the amount of data and more data in the way of training samples. The second limiting factor was to maintain balanced data. The preparation of the data was performed based on large scan fragmentation of training samples—for instance, base scans of size 640 × 11,000 were fragmented to smaller samples such as 640 × 96, etc. This way, it was possible to use stepping in fragmentation, where every sample of a specific size was extracted from a basic scan in the specific step. In this case, worse results were obtained due to unbalancing the training data, which have high impact on the training. For this reason, a dataset without stepping fragmentation was used. Another outcome was that the accuracy and success of this method were highly sensitive to the type of data. Increasing this success was based on the training and application of point clouds representing a planar surface parallel to the base XY plane. For inspection systems, it is necessary to manage more data with captured defects and to demonstrate the ability to capture defects.

## Figures and Tables

**Figure 1 sensors-23-04772-f001:**
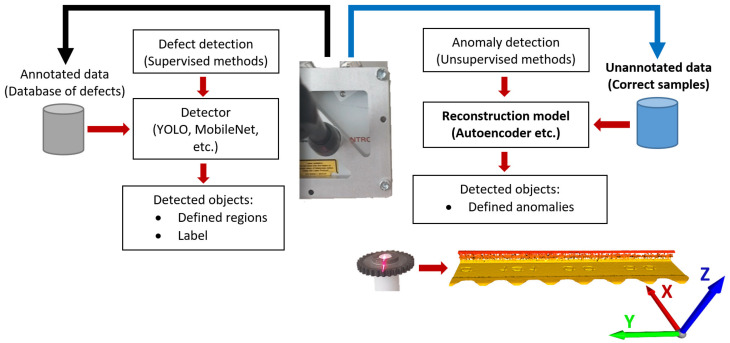
Illustration of inspection systems for defect detection separated into two main types: the defect detection by supervised methods—detectors; and the anomaly detection by unsupervised methods—mainly autoencoders.

**Figure 2 sensors-23-04772-f002:**
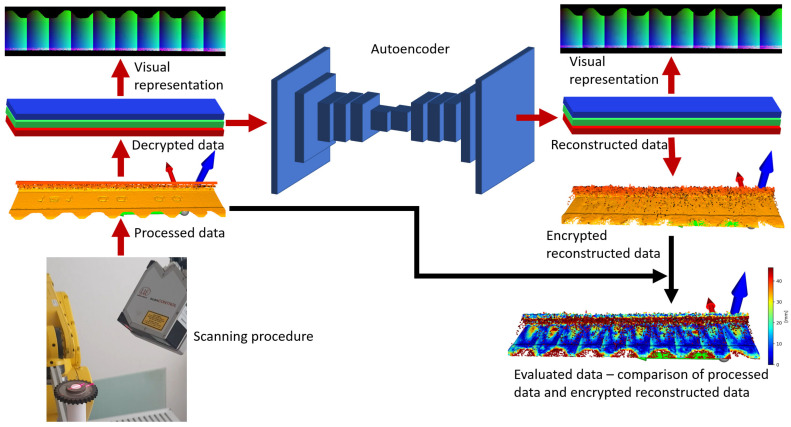
Illustration of a system consisting of obtaining data from laser sensors in scanning procedure, processing the data (point cloud), decrypting the data for autoencoder, reconstructing the data, encrypting them to point cloud, comparing the processed data (tested data), and encrypting the reconstructed data (after training of autoencoder).

**Figure 3 sensors-23-04772-f003:**
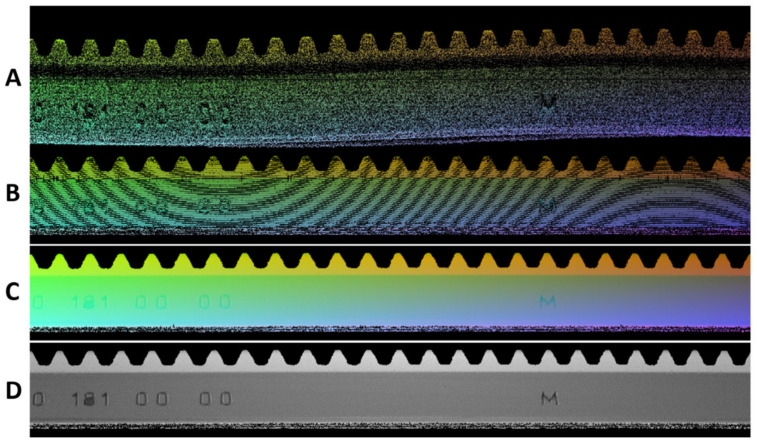
Illustration of processing of point cloud and transformation to visual content. (**A**)—basic data including the first harmonic component in the data (eccentricity during the scanning); (**B**)—removal of the first harmonic component in the data; (**C**)—filling the missing points in the matrix; (**D**)—visualization of Z-coordinates of points in grayscale.

**Figure 4 sensors-23-04772-f004:**
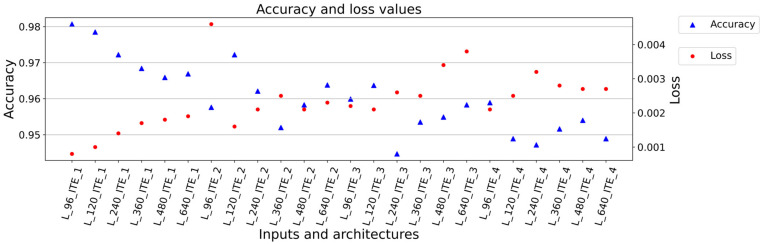
Summarizing training architectures according to Table A1, where ITE_X represents type of architecture, and L_XXX represents second size of training data.

**Figure 5 sensors-23-04772-f005:**
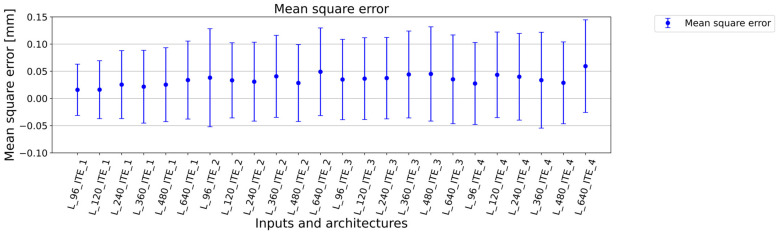
Mean square error values including standard deviation, where ITE_X represents type of architecture, and L_XXX represents second size of training data.

**Figure 6 sensors-23-04772-f006:**
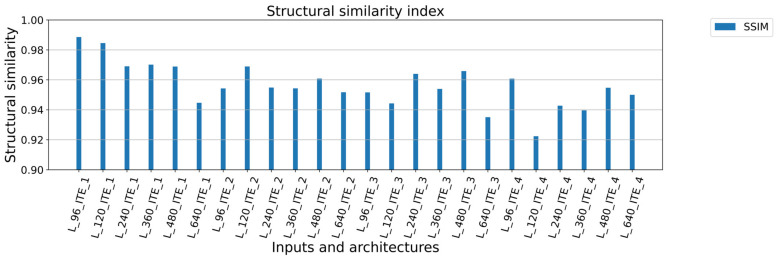
Structural similarity index (SSIM) between original samples and reconstructed samples.

**Figure 7 sensors-23-04772-f007:**
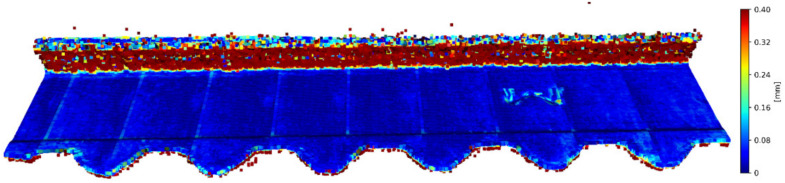
Merge of X and Y coordinates of basic points and Z coordinates from reconstructed point cloud.

**Table 1 sensors-23-04772-t001:** Summarization of used types of architecture for autoencoders (detailed described in Table A4).

Type of Architecture	No. of All Parameters	No. of Filters in One Conv. Layer	No. of Conv. Layers	Comment
ITE_1	896227	128	9	Basic architecture with small number with filters
ITE_2	7996899	384	9	The same architecture as basic architecture, but with average number of filters in convolution layers used in this work
ITE_3	26442243	512	14	Included more convolution layers, with higher number of filters in first convolution layer. The 2 convolution layers at the end of architecture in shape (None, 640, 640, 512)
ITE_4	18923011	512	10	Balanced architecture in way of similarity of convolution layers and number of parameters for start and end architecture

**Table 2 sensors-23-04772-t002:** Result summarization of architecture types.

Type of Architecture	Results
ITE_1	Basic architecture with fast training and lower consumption of GPU memory. The results are sufficient.
ITE_2	Little worse results compared to ITE_1. High sensitivity to overtraining, the necessity to use 30 epochs for training. For L_480 and L_640, 50 epochs were used for training.
ITE_3	L_96: 30 epochs, 30–50 epochs were used for other types. High sensitivity to overtraining. The results compared to other types of architectures are average. Presumably, there is a lack of data to reach better results for architectures with more convolution layers.
ITE_4	Training performed with 30 epochs. The results are below average.

**Table 3 sensors-23-04772-t003:** Error of reconstruction in specific axis.

Axis	Error in Specific Axis (L_96_ITE_1)	Mean Square Error [mm]
X axis	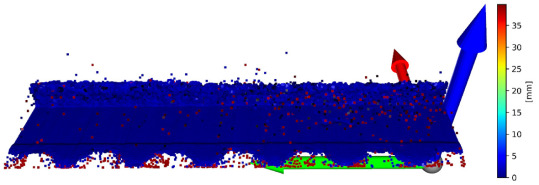	0.091116
Y axis	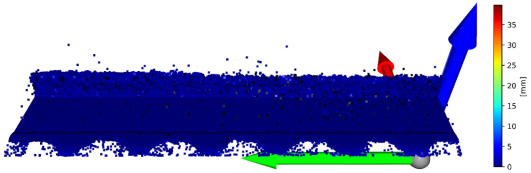	0.000000
Z axis	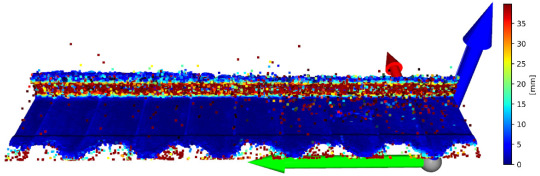	0.101825

## Data Availability

The data presented in this study are available on request from the corresponding author.

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
