# Peer review of "Autoencoders Based on 2D Convolution Implemented for Reconstruction Point Clouds from Line Laser Sensors"

_sensors, 2023, doi:10.3390/s23104772_

Round 1

Reviewer 1 Report (Previous Reviewer 1)

The article is re-presented by accepting the suggestions made. I still find the final part of the text, where the conclusions are presented in sequence, followed by a discussion, unclear. In my opinion, in general, the discussion should precede the conclusions. But in this case, I think that the two chapters could be merged into one chapter devoted to the conclusions. This would avoid some repetition. Specifically, the problems related to the characteristics of memory, underlined in lines 364, 369 and 379 could be summarised in a single sentence. 

Some minor text editing:

line 71 delete ““Type Of”

line 157 delete “work”

line 237 check “wasspecific”

Author Response

Dear reviewer.

We are very thankful to you for your suggestions and your review of this manuscript. We are really appreciating your work, which is very valuable for us, and which helped us.

Yours faithfully.

Authors.

Reviewer 2 Report (Previous Reviewer 3)

Compared to the first version of this paper, the authors have addressed some of my comments. But there seems some issues that still need to be clarified. For example, the descriptions of used datasets and the comparisons of state-of-the-arts.

The contributions can be listed at the end of the introduction for better understanding.

Moreover, some references are suggested to be added, for example, the specific point cloud evaluation methods including Blind quality assessment of 3D dense point clouds with structure guided resampling, PQA-NET, RR-CAP, etc. And the auto-encoder used in other image processing tasks, such as EAA-Net: Rethinking the Autoencoder Architecture with Intra-class Features for Medical Image Segmentation and Binocular rivalry oriented predictive autoencoding network for blind stereoscopic image quality measurement.

Please further improve the presentation of the paper, such as the size of the figures.

n/a

Author Response

Dear reviewer.

We are very thankful to you for your suggestions and your review of this manuscript. We are really appreciating your work, which is very valuable for us, and which helped us.

Yours faithfully.

Authors.

Round 2

Reviewer 2 Report (Previous Reviewer 3)

No further comments.

n/a

This manuscript is a resubmission of an earlier submission. The following is a list of the peer review reports and author responses from that submission.

Round 1

Reviewer 1 Report

The scope of the research falls outside the domain expected for the special issue, but the work could be of interest since the results are applicable in other contexts beyond the industrial one described in the article. The abstract from lines 17 to 22 describes the work division starting from the second part. In the following lines, there is a division into three parts that is not followed in the article. It is proposed to rewrite the abstract according to the publisher's guidelines.

The article is extremely difficult to read as written to the point of not allowing a reliable evaluation of the quality of the research. In general, the writing does not follow the guidelines proposed by the publisher MDPI. The introduction adequately describes the context in which the research fits, describing the research already developed and the areas to be explored. The most direct references to the proposed work are found in a separate chapter (Chapter 2) even though the topics represent a continuation of what is described in the introduction. The authors are proposed to merge the two chapters into a single introduction.

Chapter 3 describes the methods used in a too synthetic way. There is a total lack of a paragraph that describes the working environment used, the software used for the analysis. There is a total lack of description of the parameters used for evaluating the quality of the results. Some information is scattered in various chapters. The description of the workflow represented in Figure 1 is too concise. The preprocessing of the data is described adequately in paragraph 3.1. The description of the autoencoder architecture in paragraph 3.2 is almost entirely entrusted to a very long table attached in Appendix B. The authors should add a summary table within the paragraph where they highlight the main differences between the different architectures, referring to the table in the appendix only for further reading.

In Chapter 4 dedicated to results from lines 199 to 206, content is introduced that would be more properly in Chapter 3. For the description of the results, the authors refer the reader to a very long table attached in Appendix A. The authors should find a way to present a summary of the results obtained within the paragraph by highlighting the most significant ones and referring to the complete table only for further reading. The authors classify some figures as tables. There are no descriptions and comments to figures 3 and 4. There is a lack of discussion of the results, a discussion is mentioned in the conclusions from lines 265 to 269.

The work cannot be accepted in its present form

Reviewer 2 Report

Comments and suggestions for the authors:

- The title should be more consistent with the topic of the work.

- In the abstract the main results of the work should be better highlighted.

Pag 1 rows 33-35 - “Such application can be seen primarily in connection with unmanned aerial vehicles (UAVs), unmanned ground vehicles (UGV), autonomous vehicles or with household devices, for example robotic vacuum cleaners, and the like.”

Please insert references as example of your statement.

 - Pag 1  rows 40-45 - “The advantage of these line lasers is the high accuracy of scanning surfaces, where the resolution in the Z axis of these sensors ranges from 0.010 – 0.001 mm … There is a possibility to scan the inspected surface in a width of up to 10-270 mm and the number of scanned points can be up to 2048 in one line”

Please insert references to these technical details, it is not clear what kind of sensors the authors are referring to.

- Pag 1 rows 47-49 - Another type of a laser 3D device is the VR-6000…”

Please insert references to these technical details.

- The introduction to the inspection methods should be clearer, maybe a scheme should help the readers.

- Pag. 2 row 82

What is the term CASAE ?

- Pag.2 rows 86-87 - “The result can also be the integration of this data into corporate data, the cloud, etc. [13].”

The sentence is not very clear in my opinion, it should probably be rephrased.

- Pag.2  rows 87-89

The purpose and novelty of the presented work need to be better detailed.

- Pag. 2 rows 93-94 - “One of the first works in the field of work with the recognition and classification of 93 point clouds is focused on the tree solution of the neural network [14].”

The sentence should be rearranged.

- Pag.3 rows 134-135 - “Laser sensors achieve a resolution of 0.01 to 0.1 mm in the X or Y axis.”

Which sensors? insert details and technical references.

- Pag. 3 rows 136-137 - “The resolution in the Z axis is achieved in the range of 0.002-0.020 depending on sensor type.”

What is the measurement unit? This sentence appears similar to that inserted in the introduction (Pag. 1 rows 41-42), however with different resolution values.

- Pag. 3 rows 140-141 - “The standard methods described in the articles on processing point clouds are built either by 1D or 3D convolution (conv1D or 141 conv3D) using voxelization (3D).”

Which articles are the authors referring to?- The article lacks any critical discussion of the results, which is a major obstacle to understanding the work.

- The graphs in Figures 3 and 4 should be redrawn to allow legibility of the different values.

- The bibliography needs to be revised appears to be lacking, a greater number of citations would improve the article, furthermore details are missing in some references

Reviewer 3 Report

This paper proposes an autoencoder framework for 3D point clouds reconstruction. Both quantitative and visualization results demonstrate the effectiveness of the proposed network. However, the proposed model is incremental, and experiments are limited in general. Some comments are listed as follows:

1. What are the specific designs for the point clouds reconstruction? Please clarify this point. 2. In the experiments, state-of-the-art methods are suggested to be compared. 3. The used dataset in the experiments are lacking explanations. 4. Except for the reconstructed MSE results, some perception-based metrics would be considered, such as the SSIM. 5. The organization of this paper could be improved. For example, the architecture of the adopted network should not be in the appendix. 6. Many details are missing, such as the adopted loss functions. 7. It is suggested to summarize the main contributions at the end of the introduction part.